# Cholinergic network modulation in disinhibited eating behavior
Swen Hesse [1,2] ✉, Michael Rullmann [1,2], Tilman Günnewig[1,2], Eva Schweickert de Palma[1,2], Lara Burmeister [1,3], Max van Grinsven [1,3], Franziska Zientek [1,2], Julia Luthardt[1], Mohammed K. Hankir [4], Philipp M. Meyer [1], Georg-Alexander Becker [1], Marianne Patt [5], Peter Brust [6,7], Burkhard Pleger [8], Michael Stumvoll[9], Anja Hilbert [10], Matthias Blüher [9,11] & Osama Sabri [1]

Cholinergic modulation of brain reward circuitry appears to play a crucial role in information processing about salience as a key biological mechanism in obesity. However, changes in acetylcholine transmission leading to abnormal eating behavior have not been demonstrated in vivo in human obesity. Using simultaneous positron emission tomography and functional magnetic resonance imaging, we found an increased α4β2* nicotinic acetylcholine receptors (nAChR) availability in response to visually salient food cues in twenty individuals with obesity, particularly in those with high disinhibited eating behavior, whereas there was no change in sixteen volunteers served as normal weight control. This increase was accompanied by a shift from dorsal attention network activation in normal-weight controls to salience network activation in individuals with obesity indicating fundamental differences in sensory cue detection. These data should encourage further investigations into α4β2* nAChR in obesity, particularly with regard to treatment with nicotinic receptor agonists for weight loss targeting hedonic overeating.

The hedonic properties of food can drive eating behavior even when energy requirements have been met, contributing to weight gain and obesity. Such behavior is determined by the response of brain regions which are involved in conditioned rewarding cues and prefrontal inhibitory control over appetitive regions[1]. Although positron emission tomography (PET) studies have shown that susceptibility to food rewards and obesity is correlated with changes of dopamine levels and reduced availability of striatal dopamine D2 receptors in humans[2], appetitive motivational states may widely depend on the function of different striatal neuromodulators.

An alternative, not yet well tested hypothesis on disinhibited eating behavior in obesity assumes changes of the brain nicotinic acetylcholine receptor (nAChR). Focus on this receptor is based on the fact that nicotine strongly interferes with rewarding effects of food[3,4], there are lower rates of nicotine abuse in individuals with obesity[5] while high restrained eating is associated with elevated rates of smoking compared to the general population[6], and that increased intake of foods high in fat and sugar activates reward circuitries similar to those activated by smoking[7]. On the other hand, nicotine increases the preference for immediate, smaller compared with later, larger food rewards (a model for impulsive choice) due to reward insensitivity[8] while functional magnetic resonance imaging (fMRI) in

humans suggested that nicotine enhances modulation of food-cue reactivity in the ventromedial prefrontal cortex (PFC)[9].

Nicotininc AChRs are widely expressed in neuronal networks of food consumption and energy homeostasis including those relevant for the evaluation of palatability and its consequences of reward processes[10,11]; it is, however, unknown, how nAChRs specifically mediates nicotine-enhancing effects on food reinforcers. While nicotine decreases food intake through activation of hypothalamic feeding circuits (which themselves may regulate reward and contribute to the loss of control over intake thereby influence weight gain)[11,12], palatable foods activate ascending dopaminergic neurons projecting from the ventral tegmental area (VTA) to the nucleus accumbens (NAc), a key component of the reward system[13]. This is mediated by nAChRs containing β2 subunits (α4β2* and α6β2*) expressed at the somatodendritic or nerve cell terminals[14] as salient cues induce ACh release in the VTA[15]. A specific role of α4β2* nAChRs in appetite control is supported by the findings showing that stimulation of α4β2* channels within the VTA by nicotine modulates food-motivated behavior and impulsivity; α4β2* agonists such as varenicline or ABT-089 may dampen this property[16]. Thus, long-term consumption of varenicline significantly reduces sucrose consumption while long-term sucrose consumption increases α4β2* and decreases α6β2* nAChRs in the NAc[17]. Similar results were observed with cytisine, another

---

nAChR drug[17]. Moreover, the novelty seeking behaviors induced by dietary fat is mediated by increased density of β2* nAChRs in PFC and midbrain regions associated with impulsivity[18] whereas ACh release in the hippocampus and PFC affects inhibitory feedback to the amygdala, hypothalamus, and thus influences disinhibited, impulsive behavior[19,20].

Preclinical studies found β2* nAChRs in the PFC linked with poor top-down control over attention in individuals who have the propensity to attribute high incentive salience to reward cues rendering such individuals vulnerable to obesity[21]. Furthermore, tonically active cholinergic inter-neurons are synchronized across wide areas of the striatum with inputs from thalamostriatal projections and access to energy-sensing systems of the hypothalamus[22]. During reward learning, the arrival of a conditioned stimulus evokes a pause response in the firing of these interneurons, whereas the same stimulus triggers an increase in the firing rate of the DA neurons in the VTA/substantia nigra (SN). The thalamically evoked pause depends upon nicotinic stimulation of DA terminals and activation of D2R expressed by cholinergic interneurons thus contributing to the reinforcing properties of food and compulsive food intake[23]. Such mechanism leads to an attentional shift following the salient stimulus whereas the nAChR antagonist, mecamylamine can diminish the thalamically evoked pause[24].

In sum, ACh coordinates the response of neuronal networks in many brain areas relevant for appetite control suggesting that α4β2* nAChR-mediated modulation is a crucial mechanism underlying eating behaviors. This primarily involves the allocation of attention and effects toward food rewards. Beyond the dopamine system, overeating may be caused by a dysfunction of modulatory nAChRs and associated failures in dietary restraints. Thus, the identification of the α4β2* nAChR as a molecular target and regulatory node in the brain networks for appetite suppression is a step towards finding healthy alternatives to smoking for weight control as a value for people with obesity.

This first in vivo study in humans investigates the behavior of central α4β2* nAChRs under rest and under visual food cue stimulation. We hypothesized that thalamic α4β2* nAChR availability under stimulation is higher in individuals with obesity, specifically with high-disinhibited eating behavior compared with normal-weight healthy individuals.

## Results

### Volumes of distribution of α4β2* nAChR

To determine α4β2* nAChR availability, α4β2* selective (-)-[^18^F]Flubatine[25] was applied together with simultaneous PET/magnetic resonance imaging (MRI) using a bolus-infusion protocol[26] in 20 participants with obesity (body mass index BMI ≥ 30 kg/m²) of high- (n = 12, BMI 38.3 ± 3.0 kg/m²; 10 female, age 34.3 ± 11.8 years, 20–55 years) or low (n = 8; BMI 37.7 ± 2.7 kg/m², 4 female, 42.6 ± 16.5, 23–62 years) disinhibited eating behavior compared with 16 normal-weight controls (BMI 18–25 kg/m²) with low-disinhibited eating behavior (n = 16; BMI 21.8 ± 1.9 kg/m², 14 females; age 27.5 ± 7.4 years, 19–45 years). Disinhibited eating behavior stats was determined using the disinhibition score of the German version of the Three-Factor Eating Questionnaire (TFEQ) with a cut off >7[27] (see Methods). All participants were scanned twice, first under baseline condition at rest, second under a visual food cue stimulation to test for changes that are mediated by α4β2* nAChR (Fig. 1a).

Overall, we found lower volumes of distribution ($V_T$) in participants with obesity as compared with normal-weight controls (pooled resting state and stimulus values) in the VTA (17.2 ± 2.1 versus 15.7 ± 1.8; $F_{(1,25)}$ = 5.180, p = 0.032; p = 0.00647 corrected for multiple comparison) and in nucleus basalis of Meynert (NBM), the primary source of cholinergic innervation to the cortex (11.1 ± 1.1 versus 9.8 ± 1.5; $F_{(1,25)}$ = 5.99, p = 0.022; p = 0.0037 corrected for multiple comparison) but not in the other regions-of-interest (amygdala 8.7 ± 0.8 versus 8.7 ± 0.8; $F_{(1,25)}$ = 0.017, p = 0.896; hypothalamus 13.1 ± 1.2 versus 12.7 ± 1.9; $F_{(1,25)}$ = 0.097, p = 0.785; insula 9.3 ± 0.8 versus 9.2 ± 0.8; $F_{(1,25)}$ = 0.429, p = 0.519; NAc 10.9 ± 0.9 versus 10.2 ± 1.0; $F_{(1,25)}$ = 0.164, p = 0.689; PFC 8.5 ± 1.0 versus 8.6 ± 0.8; $F_{(1,25)}$ = 0.336, p = 0.567; thalamus 34.3 ± 5.0 versus 35.7 ± 5.0 $F_{(1,25)}$ = 0.741, p = 0.398); cerebellum 11.5 ± 1.4 versus 11.8 ± 1.2 $F_{(1,25)}$ = 1.244, p = 0.275) (Fig. 1b).

In order to identity between-group effects in all the three groups considering disinhibited eating behavior for the resting state and for the stimulus condition, we applied ANCOVA with age as a covariate. We found significantly higher $V_T$ values under stimulation in participants with obesity as compared to normal-weight controls in the thalamus (37.0 ± 5.5 versus 33.5 ± 4.7 versus MW ± SD; $F_{(1,27)}$ = 5.10, p = 0.026) while no other statistically significant effects were depicted (amygdala $F_{(1,29)}$ = 0.188; p = 0.688; hypothalamus $F_{(1,29)}$ = 0.429; p = 0.517; insula $F_{(1,29)}$ = 0.009; p = 0.924; NAc $F_{(1,29)}$ = 0.332; p = 0.569; PFC $F_{(1,29)}$ = 0.012; p = 0.914; thalamus $F_{(1,29)}$ = 0.265; p = 0.610; VTA $F_{(1,29)}$ = 3.269; p = 0.081; NBM $F_{(1,29)}$ = 5.674; p = 0.024; cerebellum $F_{(1,29)}$ = 0.008; p = 0.183); stimulus: amygdala $F_{(1,27)}$ = 0.028; p = 0.869; hypothalamus $F_{(1,27)}$ < 0.001; p = 0.990; insula $F_{(1,27)}$ = 1.507; p = 0.230; NAc $F_{(1,27)}$ = 0.281; p = 0.600; PFC $F_{(1,27)}$ = 0.890; p = 0.354; thalamus; VTA $F_{(1,27)}$ = 0.991; p = 0.081; NBM $F_{(1,27)}$ = 3.475; p = 0.073; cerebellum $F_{(1,27)}$ = 1.866; p = 0.183) (Fig. 1b, c). Within each subgroup, $V_T$ increased during the stimulus condition compared with resting state examination in individuals with obesity and high-disinhibited eating behavior in the thalamus with borderline significance 33.4 ± 3.8; $T_{(17)}$ = -2.06, p = 0.05469; individuals with low disinhibited eating behavior $T_{(11)}$ = 0.13, p = 0.8988; normal-weight controls $T_{(28)}$ = 0.945, p = 0.3527; for regional correlation between $V_T$ at rest and $V_T$ under visual food cue stimulation in normal-weight controls and obesity see Supplementary Fig. 1).

### Neuronal network connectivity in resting-state versus visual food cue stimulation

To assess brain wide, network level of information processing, we performed an MRI-based function connectivity analysis comparing resting and stimulus conditions. Using the thalamus[28] as a seed in these joint PET-MRI analyses on functional connectivity, we found strengthened connectivity with the posterior and central part of the brain in normal-weight controls while there was a strengthened connectivity with the insula on both sides in obesity (Fig. 2). This change indicates a shift from attentional network activation in normal-weights towards salience network activation in individuals with obesity when presented with visual food cues. Using the NBM as a seed, we found strengthened connectivity to VTA in normal-win normal-weight controls (resting state versus visual food cue stimulus), which was not the case in individuals with obesity (Supplementary Fig. 3). In individuals with obesity, the VTA was strongly connected to the ventrolateral prefrontal cortex next to the insula bilaterally (Supplementary Fig. 3).

### Regional association between α4β2* nAChR availability and network strength under visual food cue stimulation

To determine regional-specific association between α4β2* nAChR availability and neuronal network activity, we extracted the mean beta estimates of the seed-based functional connectivity MRI analysis of those clusters resulting from the SPM comparison between resting state and food-cue stimulation in individuals with obesity and in normal-weight controls. According to our hypothesis, we first tested whether the changes in $V_T$ observed in individuals with obesity are related to the beta estimates in each mask. This analysis showed significant relationship between $V_T$ in the thalamus and beta estimates in the network changes in obesity but not in normal-weight controls (Fig. 3a). We then applied the same mask to the parametric images of α4β2* nAChR availability to test exploratory whether individual $V_T$ values and beta estimates are related in non-thalamic regions in both networks that showed significant differences between rest and visual food cue stimulation. We found a significant positive correlation between $V_T$ and beta estimates in obesity, which was not the case when applying the same mask to the group of individuals with normal-weight (Supplementary Fig. 4a–d).

### Relationship between α4β2* nAChR availability, neuronal activity and eating behavior

To stratify whether study participants have high or low disinhibited eating, we used the TFEQ as a long-term trait attitude of dysregulated eating behavior. We did not find statistically significant associative patterns related

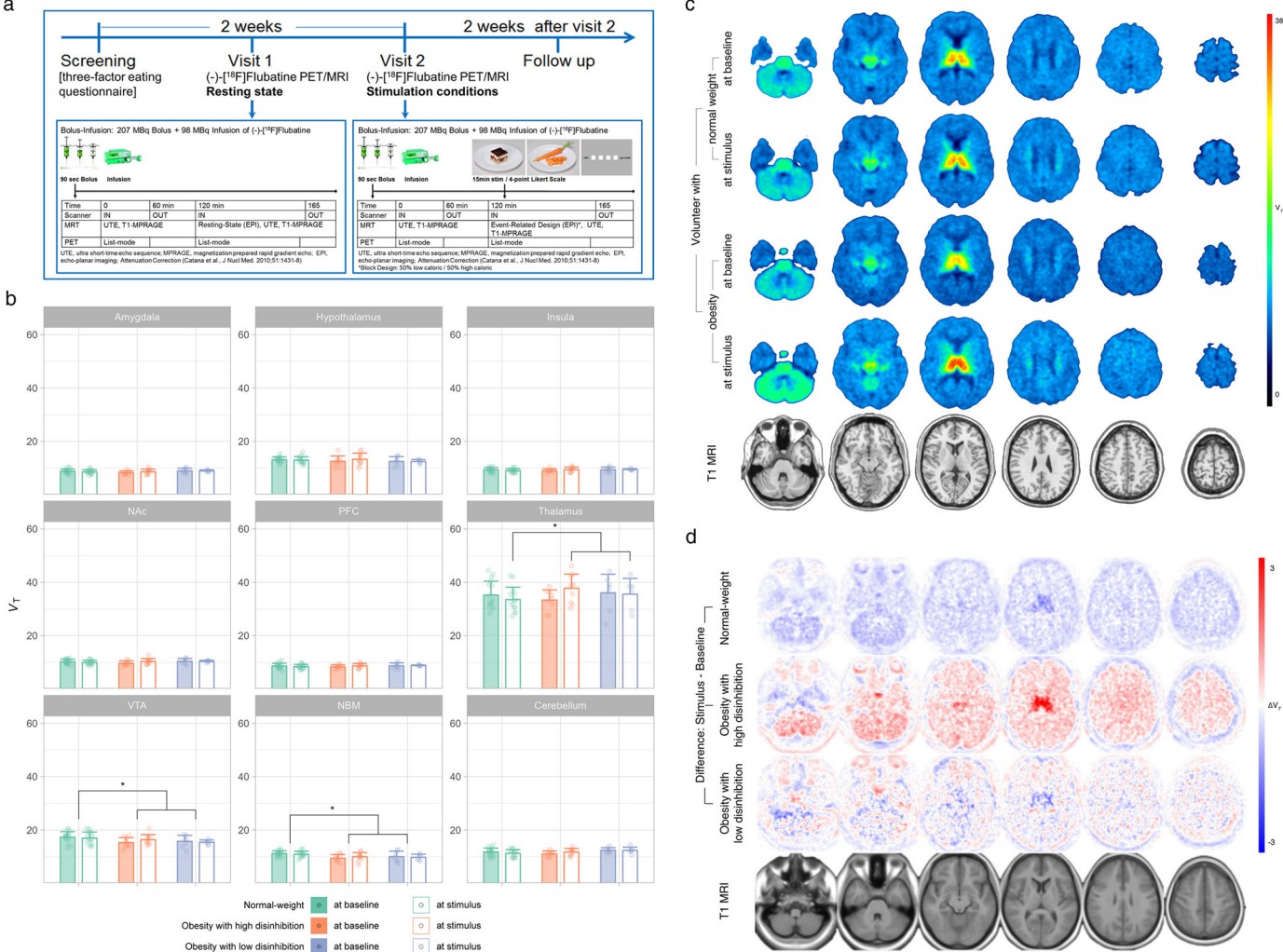

**Fig. 1 | Visual food cue stimulation induces an increase in α4β2\* nicotinic acetylcholine receptor availability in individuals with obesity and disinhibited eating behavior. a** Visual food cue paradigm during scan compared with resting state examination (see also Methods) **b** Volumes of distribution ($V_T$) in all study participants of the three groups indicating significantly lower $V_T$ ($p < 0.05$) participants with obesity compared with normal-weight controls regardless whether they were at resting state or under stimulation. The presentation of food cues itself led to significant between-subject effects with elevated $V_T$ in the thalamus in individuals with obesity and high disinhibited eating behavior ($p < 0.05$). **c** Representative

parametric images of $V_T$ scaled to the same maximum indicating an increase of $V_T$ in one individual with obesity under visual food-cue stimulation as compared with one normal-weight healthy control. For individuals $V_T$ changes between rest and stimulus see Supplementary Fig. 2. **d** Parametric images of averaged $V_T$ changes of the observed differences between rest and stimulus for normal-weight healthy controls, individuals with obesity with low and high disinhibited eating behavior (stereotactically normalized into MNI space). Compared to rest condition, blue colors indicate a decrease and red colors an increase at stimulus condition.

to α4β2\* nAChR availability at resting state albeit the correlation between $V_T$ in the thalamus and TFEQ 'cognitive restraint' was different between normal-weight controls and individuals with obesity ($Z = 2.28$, \*$p = 0.01$) (Supplementary Fig. 5). In order to assess the relationship between changes of α4β2\* nAChR availability and immediate changes of eating behavior, a continuous visual analog scale (VAS) for self-rated hunger, wanting, liking, and disinhibition in the current context of visual food cue presentation ranging from 0 (=not at all) to 100 (=extremely) was applied (see Methods and Supplementary Fig. 6). Neither changes in $V_T$ nor beta estimates correlated with changes from the subjective valuation before and after resting state to the self-reported feeling before and after visual food cue stimulation (Fig. 4). The changes in $V_T$ were not predicted by the TFEQ score overall while beta estimates showed association with the BMI in individuals with obesity regardless of whether we use the thalamus or the VTA as seed region (Supplementary Fig. 6). However, the changes of α4β2\* nAChR availability either in the thalamus or the VTA depend on TFEQ 'cognitive restraint' score, that is the higher the score the higher the changes in $V_T$ in individuals with obesity (Fig. 4, Supplementary Fig. 5, Supplementary Fig. 8). Although

VAS scores for disinhibition significantly differed between the three groups (normal-weight controls, individuals with obesity and low disinhibited eating behavior, individuals with obesity and high disinhibited eating behavior) before ($12.8 \pm 16.2$, $31.1 \pm 23.9$, $34.7 \pm 26.0$, $p = 0.002$) and after scanning ($12.9 \pm 18.2$, $25.6 \pm 21.6$, $38.4 \pm 28.5$, $p = 0.002$; $p = 0.006$ comparing scores and in individuals with high disinhibited eating behavior before and after resting state), we did not observe an association between changes of either $V_T$ or beta estimates. (Supplementary Fig. 8–9).

## Discussion
The global obesity epidemic poses a major challenge for health care systems worldwide. Thus, the search for interventions to achieve sustainable weight loss reaches high priority including a thorough investigation of principle biological and behavioral mechanisms in individuals with obesity. As a key biological mechanism in obesity and putative pharmacological treatment target, the brain cholinergic system gains interest since cholinergic modulation of brain reward and attentional networks seems to play a crucial role in information processing about salience. Here, we investigated changes in

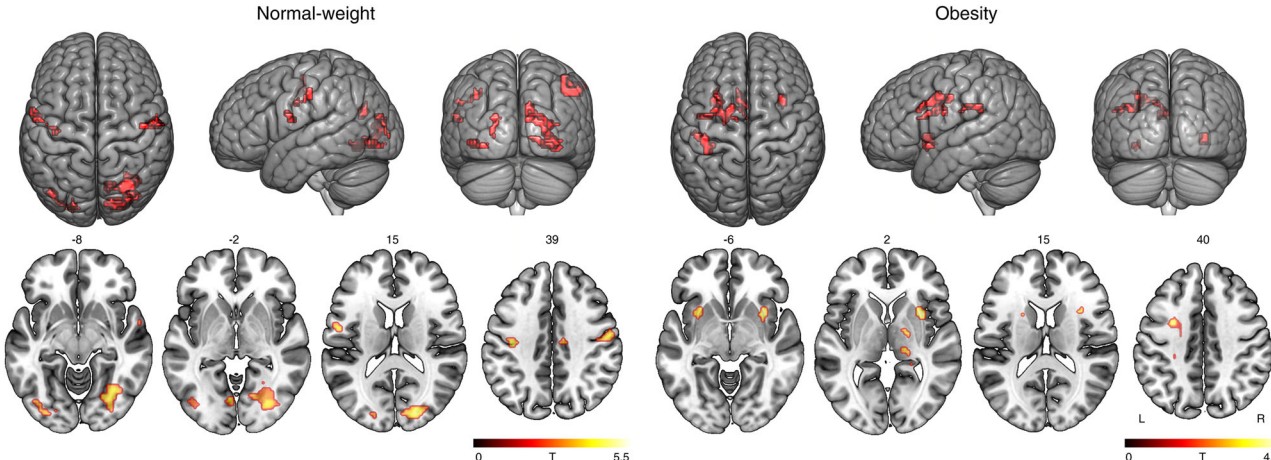

**Fig. 2 | Functional connectivity under stimulation versus resting state in individuals with normal-weight and in individuals with obesity indicating a shift from attentional to salience network activation when centering the thalamus.** Statistical parametric mapping of resting state versus food-cue stimulation data using the thalamus as the seed region and the Hillmer-corrected thalamic $V_T$ as a covariate ($p_{uncorrected} < 0.001$; two sampled $t$ test, Rest < Food, unpaired, $n = 16$ resting state examinations versus 14 investigations under visual food cue stimulation in normal-weight healthy controls; $n = 16$ individuals with obesity). The paired $t$ test (Rest < Food) yielded to a similar pattern with more laterality to the left hemisphere (Supplementary Fig. 3).

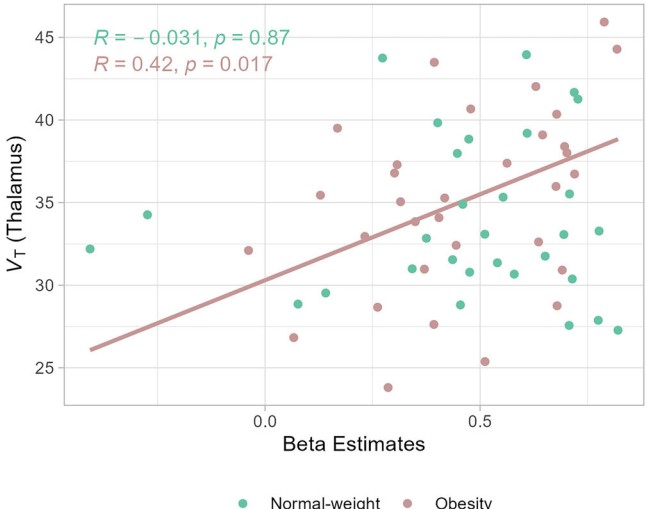

**Fig. 3 | α4β2* nAChR availability and neuronal network signals are regional-specific associated in individuals with obesity.** Individual distribution volumes $V_T$ of the thalamus and extracted blood-oxygen level dependent (BOLD) fMRI beta estimates from clusters resulting from the SPM comparison between resting state and food-cue stimulation in either individuals with obesity or in normal-weight controls ($p < 0.001$, $k > 10$ voxel, Fig. 2) are significantly correlated in individuals with obesity (Pearson's $r = 0.42$, $p = 0.017$) but not in normal-weight controls (Pearson's $r = -0.031$, $p = 0.87$).

α4β2* nAChRs in response to salient food cues together with assessments of changes in large-scale networks and behavioral ratings in individuals with obesity with varying levels of disinhibited eating behavior and normal-weight controls.

In the present study, we demonstrated for the first time in vivo neural changes under visual food cue stimulation in people with obesity together with changes of α4β2* nAChR in key areas of the brain mediating eating behavior. These data together provide fundamental insight into the mechanisms encoding motivational salience processing as they indicate immediate responses of α4β2* nAChR and neuronal networks following cue presentation. This includes the engagement of several brain regions including homeostatic as well as hedonic circuits. We saw an increase in α4β2* nAChR availability predominantly in the thalamus when comparing groups of normal-weight controls and individuals with obesity and high disinhibited eating behavior. There was a similar increase in α4β2* nAChR availability in the VTA and the NBM when comparing resting-state and stimulation in obesity (while α4β2* nAChR availability stayed unchanged in

normal-weight controls). However, the biological explanation of this findings is difficult as many processes - from trafficking, desensitization and internalization to compensatory upregulation - can influence nAChR availability[29,30]. As these processes have not been well investigated in human in vivo studies so far, we would like to propose a possible mechanism based on previous modeling data considering the ability of ACh to coordinate the response of neuronal networks in many brain areas that makes cholinergic modulation an essential mechanism underlying complex behaviors[15,31].

Nicotinic AChRs belong to a super-family of Cysloop ligand-gated ion channels that respond to endogenous ACh or other cholinergic ligands[32]. The nAChR is a type of allosteric receptor that responds with an increase in available sensitive binding sites after phasic AChR release though. It is currently not possible to distinguish whether higher radioligand binding marks the high-affine desensitized state or the open receptor state where nAChR exerts its excitatory function[31]. Research in smokers showed that chronic tobacco smoking leads to a rapid nicotine-induced upregulation of nAChRs as well as profound changes in conformational states[33]. This results

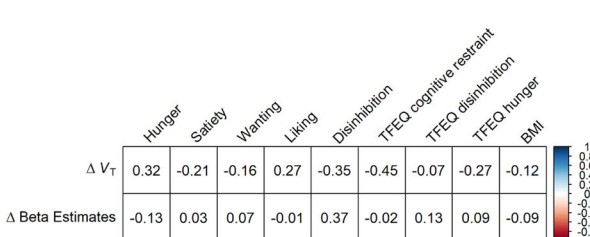

**Fig. 4 | Group- and condition-specific correlation coefficients (Spearman rank correlation) between changes of visual analog scale (VAS) assessments, $V_T$ and fMRI beta estimates as well as TFEQ in. a** normal-weight controls and **b** individuals with obesity ($\Delta$VAS = VAS$_{post-pre(stim)}$ - VAS$_{post-pre(rest)}$). fMRI beta estimates

showed correlation with changes in satiety ($p = 0.0196$) in individuals with obesity and with BMI ($p = 0.007$) while changes in the $V_T$ of the thalamus were significantly associated with the TFEQ 'cognitive restraint' ($p = 0.0152$). $V_T$ and fMRI estimates are extracted from the clusters as presented in obesity in Fig. 3.

in desensitized nAChRs thereby disrupting endogenous ACh signaling, which is pivotal for computing environmental salient cues, e.g., visual food stimuli. Preclinically, cholinergic transients (resulting from the hydrolysis of newly released ACh) appear to be essential for synchronized output driven by salient cues on multiple times scales[34,35]. While fast cholinergic signaling mediates bottom-up, or cue-driven attention, slower cholinergic signaling influences top-down, or goal-driven attention by stabilizing task-sets and context representation[35]. The transient component appears to support the activation of cue-associated tasks or response sets. Translated to our data, normal-weight controls perform better in terms of stabilizing the task set (looking to visual food cues), presumably by maintaining the synchrony of the representative cortical ensembles with no or slow changes in nAChR availability when food is presented. This is explained by the fact that higher levels of tonic ACh are generated during task performance due in part to activation of mesolimbic-cholinergic interactions and these augmented levels of cholinergic activity are thought to amplify cue-evoked glutamate release from thalamic inputs to the cortex, thereby enhancing cortical mechanisms mediating the detection of signals (for ref. see ref. 35).

Individuals with obesity, in contrast to normal-weight controls, recruit ventral attentional salience networks in response to food cues nicely corroborating preclinical data that showed poor performance in sustained attention tasks in those rats that are prone to attribute high incentive salience to reward cues depending on α4β2* nAChR availability[21]. The increase in α4β2* nAChR availability could be interpreted as a stimulation/recruitment of cortical circuits presumably consistent with higher effort for cognitive restraint (indicated by the TFEQ) to behavior elicited by rewarding food cues since we also found a rather low α4β2* nAChR availability at baseline in those participants with obesity that exhibit high disinhibited eating behavior. This is in line with studies showing that a low-capacity choline transporter in humans is associated with distractibility and stimulus-driven behavior while the loss of cholinergic neurons broadly disrupts attention and associated timing, working memory, and movement control[35].

The cortical patterns seen in our studies are also in line with preclinical data that showed how salient stimuli redirect attention and suppress ongoing motor activity based on cholinergic controlled thalamic gating of corticostriatal signaling[24]. This fundamental regulation appears to strongly differ in individuals with obesity compared with normal-weight control recruiting different thalamico-cortical networks. We did not perform a dedicated parcellation of the thalamus; however, it is most likely that interregional communication is mainly mediated by the mediodorsal thalamus[28,36]. An alternative hypothesis is that there is a shift between the thalamic hubs, for example from anteriomedial nuclei that project to frontoparietal regions to the mediodorsal thalamus[37] while α4β2* nAChR contributes to a differential modulation of ACh on the paraventricular thalamus, a central relay station connecting brainstem and hypothalamic signals with the limbic forebrain in emotional and motivational contexts[38].

Since α4β2* nAChRs do not downregulate as a result of continuous stimulation, these receptors are a promising target for agonist treatment. Indeed α4β2* nAChR agonists, including ABT-089, enhance attentional

control (for ref. see ref. 35) in susceptible individuals who otherwise have difficulties with suppressing attention to reward cues due to altered cholinergic transmission in thalamic and mesolimbic areas.

Stimulation also induced strengthened connectivity between the VTA and the ventrolateral PFC, which is a key region for evaluating the emotional significance of external stimuli, and also serves as an important substrate for cognitive influences on emotional states[39]. This mechanism, together with salience network activation from the thalamus including the insula, may contribute to the value computation in situations of high arousal or attention such as the presence of food[40]. On the other hand, the strengthened connectivity between the NBM and the VTA found in normal-weight controls under rest was absent in obesity pointing to the fact that both local neural ACh circuits and extended brain networks are involved in the modulation of salient food cue processing. This finding is also consistent with the topographic, rather than diffuse, organization of basal forebrain cholinergic neurons and their outputs[41,42].

The relationship between cerebellar α4β2* nAChR availability and cognitive control we found in individuals with normal weight but not individuals with obesity in the present study are in line with previous work on patients with Prader-Willi syndrome (PWS), a genetic form of obesity characterized by disinhibited eating[43]. It was shown that PWS patients had reduced deep cerebellar functional responses to food images in the fasted state compared to individuals with normal weight[43]. Accordingly, a glutamatergic population of neurons in the anterior deep cerebellar nucleus of mice was found to suppress food intake by modulating dopaminergic VTA neurons which project to the NAc[43]. These findings, along with those of the present study, suggest that cerebellar neurons serve to enhance cognitive control of food intake, and that this effect is diminished in obesity. However, we did not find a specific relationship between the imaging parameters, in particular their changes between resting state and stimulation, and the state of eating behavior as indicated by VAS.

Our data are also in line with a dopaminergic pathway from the VTA to the PFC which is crucial for the rewarding and motivational properties of food and presumably non-dopaminergic mechanism with glutamatergic input to the VTA and a GABA-ergic output from the VTA to the pedunculopontine tegmental nucleus[44]. Blocking the α4β2* nAChR after smoking by partial agonists has revealed that the degree of receptor occupancy directly correlates with the extent of dopamine release[45]. Nevertheless, it remains to be shown whether partial agonists such as varenicline would reduce the reinforcing properties of food in obesity by altering dopamine release to reinstall normal circuitry function and normalize eating behavior.

## Methods
### Study participants
The study was performed in accordance with the Declaration of Helsinki with the guidelines for Good Clinical Practice (GCP), approved by the local ethics committee of the Medical Faculty of the University of Leipzig and the *Bundesamt für Strahlenschutz* (BfS; Federal Office for Radiation Protection), and registered under

DRKS00010927 at the *Deutsches Register für klinische Studien* (DRKS; German Registry for Clinical Studies). The study participants were recruited during ecotrophological consults at the University Hospital or by using public postings. Inclusion criteria (at Screening visit V0) included two sexes (male and female), a BMI above 30 kg/m$^2$ (for one of the obesity groups) or below 25 kg/m$^2$ (for the normal-weight control group) and had a body shape that fitted in the scanner, an age between 18 and 65 years and provided written informed consent. To ensure that there were no influences from smoking on the α4β2* nAChR availability measures, participants had to be non-smokers for a period of at least 6 months. The exclusion criteria were contraindications for nuclear imaging or MRI imaging (e.g., pregnancy (excluded by β human chorionic gonadotropin) or breastfeeding, claustrophobia, pacemakers, or ferromagnetic devices); neuropsychiatric or neurological disorders; previous neurosurgical operations; structural brain lesions; the use of medication for weight reduction in the past 6 month or bariatric surgery; alcohol and substance abuse; and vegan diet. To exclude eating disorders or manifest depression, we applied Eating Disorder Examination Questionnaire (EDE-Q8)[46] and Beck Depression Inventory (BDI II)[47], respectively. The German version of the Three-Factor Eating Questionnaire (TFEQ) covers three domains of eating behavior, which are 'cognitive restraint', 'disinhibition' and 'hunger')[48].

### PET/MR imaging
The radiotracer (-)-[$^{18}$F]Flubatine possesses fast brain kinetics and favorable imaging properties which makes the tracer highly suitable for in vivo imaging of α4β2* nAChR[25]. All participants underwent two visits during which the (-)-[$^{18}$F]Flubatine PET/MRI measures took place (visit 1, resting state examination; visit 2, visual food cue stimulation). Before each scan, all participants underwent a drug screening to ensure that no other substances would interfere with the imaging data. Female participants furthermore had to do a pregnancy test. The radioligand was synthesized using the synthesis module TRACERlab FX FN (GE Medical systems) under good manufacturing practice conditions with high radiochemical yield, purity, and high specific activity[49]. Within the bolus-infusion protocol[26], a total of 300 MBq of (-)-[$^{18}$F]Flubatine was applied with 207 MBq of (-)-[$^{18}$F]Flubatine being applied during a bolus injection within a period of 90 seconds and 98 MBq of (-)-[$^{18}$F]Flubatine being subsequently applied during a constant tracer infusion with an infusion rate of 40 ml/h. The scans were conducted using the Biograph mMR scanner (Siemens Healthineers, Erlangen). Each scan lasted 165 min in total which was divided into two blocks. The first block started at the moment of tracer injection and lasted 60 min. After that, participants continued to receive radiotracer infusion outside of the scanner to reach an equilibrium after 120 min post injection. During the second block, both fMRI data using echo-planar imaging and structural MR imaging data (using T1-weighted MP-RAGE) were acquired alongside three-dimensional PET data acquisition with the following parameters: MP-RAGE (176 contiguous sagittal slices with 1 mm thickness and no gap; repetition time (TR)/echo time (TE) = 1900/2.53 ms; inversion time (TI) = 900 ms; flip angle = 9°; field of view = 250 × 250 mm; matrix = 512 × 512; voxel size = 1.0 × 0.48 × 0.48 mm) and BOLD fMRI (600 echo planar imaging (EPI) volumes with a voxel size of 3 × 3 × 4.2 mm, TR = 2000 ms, TE = 30 ms, flip angle = 90°, and slice thickness of 3.5 mm).

### Individual scan day
Study participants were ask to have a light breakfast before arriving at the PET unit in the morning at 8:00 am. The scan starts at 10:00 am and participants were asked about their feelings of hunger and satiety immediately beforehand using the VAS, which did not differ in the hunger sub-score between the groups with mean values of 16.4 ± 15.8 in normal-weight controls, 19.3 ± 13.9 in people with obesity and low disinhibited eating behavior and 15.7 ± 14.2 in people with obesity and high disinhibited eating behavior ($p = 0.16$).

### Visual food cue stimulation
Food cues[50] (Supplementary Fig. 6) were presented during the 2nd scan once an equilibrium was reached after 120 min post injection. For the presentation of food cues, a video projection system presented food pictures (Presentation, NeuroBehavioral Systems, Inc., Berkeley, USA) via a projector using a screen viewed through a mirror in the head coil. During the task, a set of visual food cues ($n = 80$) was presented in a randomized block design (high- and low-caloric food) with each cue being presented for a period of 3 s. A randomized jitter (between 1.8 and 7.8 s) was applied between each cue. Next to and after each scan, a visual analogue scale (VAS) was used to obtain data on feelings of hunger, wanting, disinhibition, satiety, and taste. Participants were asked to rate these factors on a continuous scale which was later translated into numbers ranging from 0 ( = not at all) to 100 ( = extremely) by measuring the distance in mm with a precision ruler (Supplementary Fig. 7). About two weeks after the second scan, participants received a follow-up via telephone.

### Imaging data processing and metabolites
Co-registration and motion-correction procedures of PET and MR imaging data were performed using SPM12-software (*Statistical Parametrical Mapping, Wellcome Trust Centre for Neuroimaging, University College London, UK*). Volumes of interest (VOIs), which were selected according to their role in central cholinergic transmission, either as source regions or as downstream regions with important modulatory function within attention and reward circuits and networks, were manually drawn in five consecutive transversal brain slices using PMOD software (*PMOD Technologies Ltd., Zurich, Switzerland*). Left and right VOIs of each hemisphere were summed to their mean value to conduct statistical analyses). Venous blood sampling was obtained for the calculation of $V_T$ at 90, 105, 120, 135, 150, 165 min p. i. This was based on the 120–165 min p. i. tracer concentration in tissue ($C_{tissue}$) at equilibrium divided by total radioactivity concentration in venous blood plasma ($C_{plasma}$), according to Innis et al.[51]. Each sampling included 4 serum tubes which contained 2 ml, adding up to a total of 48 ml. We applied a tissue clearance correction in VOIs (i.e., the thalamus, VTA, NBM) that do not reach true equilibrium after 120 min p. i. to reduce the bias in $V_T$ estimates[52].

### MRI data analysis
Resting-state functional magnetic resonance imaging (rs-fMRI) data were preprocessed and analyzed using the RESTplus toolbox (v1.30), a MATLAB-based tool designed for resting-state fMRI analysis[53]. Basic pre-processing steps included the removal of the first 10 time points, motion correction, slice timing correction, realignment, reorientation, normalization, and detrending. For the seed-based fMRI analysis we build a general linear model on single-subject level. After applying the seed to extract the first Eigenvariate of the beta values across all voxels within the seed mask, the resulting individual time series was implemented within the same single-subject model as an additional non-interacting regressor to test for a positive correlation (i.e., strengthened connectivity) of the seed region throughout the entire brain. The individual statistical maps were entered into a group-level two-sampled t-test with the corresponding individual $V_T$ of the seed region to assess differences in functional connectivity of the seed in relation to the individual $V_T$ (i.e., interaction between fMRI and $V_T$) between both scanning conditions, at rest and with visual stimulus. Seed masks were defined for the thalamus, VTA and NBM.

### Statistics
All analyses were carried out using Matlab (The Mathworks Inc., Natick, USA) and R. For the statistical analysis a two-way mixed ANOVA was conducted to test for between-subject effects and an ANCOVA with age as a covariate to test for between-group effects. Spearman rank correlation analyses were conducted to examine the associations between the $V_T$ values, fMRI beta estimates, TFEQ and VAS scores. Significance level was set at $p < 0.05$.

## Reporting summary

Further information on research design is available in the Nature Portfolio Reporting Summary linked to this article.

## Data availability

All source data underlying the graphs and charts are uploaded as Supplementary Data 1. All other datasets and codes are available from the corresponding author upon reasonable request.

## Code availability

All scripts and custom code are available upon request to the corresponding author.

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

## Acknowledgements

This work was supported by financial support from the Federal Ministry of Education and Research (BMBF), Germany, F.Z.: [01E01001]. We are very grateful to Jane Neumann (Department of Medical Engineering and Biotechnology, University of Applied Sciences, Jena, Germany) for first discussions on fMRI design. We deeply thank all the medical assistants, in particular Martin Wehner and Torsten Böhm without whom it would not have been possible to carry out the investigations, and all the study participants for taking part in this trial. We are also thankful to Sarah Martin, Corinna Seybold und Danielle Schewe for helping with neuropsychological data collection, and Anja Landsmann (Institute for Drug Discovery, University of Leipzig) for her outstanding administrative work within the Integrative Research and Treatment Center Adiposity Diseases Leipzig supporting this study. The data were in part presented at the 33$^{rd}$ Annual Congresses of the European Association of Nuclear Medicine, October 22–30$^{th}$, 2020 (virtual congress), the 34$^{th}$ Annual Congresses of the European Association of Nuclear Medicine, October 20–24$^{th}$, 2021 (virtual congress); at the 11$^{th}$ European Conference of Clinical Neuroimaging, Geneva (Switzerland), March 14–15$^{th}$, 2022 and at the Annual Meeting of the Society of Nuclear Medicine and Molecular Imaging, June 24–27$^{th}$, 2023, Chicago, Illinois (United States of America). Supported by the Open Access Publishing Fund of Leipzig University.

## Author contributions

S.H.: Study conception and design, data acquisition, analysis and interpretation. Drafting the article. M.R.: Study conception and design, data acquisition, analysis and interpretation. Drafting the article. T. G.: Data acquisition, analysis and interpretation. E. S. de P.: Data acquisition, analysis and interpretation. M. v. G.: Data analysis and interpretation. L.B.: Data analysis. F.Z.: Data acquisition and study coordination. J.L.: Data acquisition and data analysis. M.K.H.: Data interpretation. P.M.M.: Data acquisition. G.-A.B.: Study design, data analysis. M.P.: Data acquisition. P.B.: Study conception and design. B.P.: Study conception and design, data interpretation. M.S.: Study conception and design. A.H.: Study conception and design. M. B.: Study conception and design. O.S.: Study conception and design, data interpretation. All authors commented on previous versions of the manuscript. All authors read and approved the final manuscript.

## Funding

## Competing interests

S.H. received research grants from the German Federal Ministry of Education and Research, German Research Foundation, Innovation Fund, and Roland Ernst Foundation for Health Care; honoraria for lectures and travel grants from GE Healthcare, Bayer Pharma, and Hermes Medical Solutions. A.H. reports receiving research grants from the German Federal Ministry of Education and Research, German Research Foundation, Innovation Fund, and Roland Ernst Foundation for Health Care; royalties for books on the treatment of eating disorders and obesity with Hogrefe and Kohlhammer; honoraria for workshops and lectures on eating disorders and obesity and their treatment, including Lilly and Novo Nordisk; honoraria as editor of the International Journal of Eating Disorders; and honoraria as a consultant for Takeda. M.B. received honoraria for lectures or consultancy from Amgen, AstraZeneca, Bayer, Boehringer Ingelheim, Daiichi Sankyo, Lilly, MSS, Novo Nordisk, and Sanofi. The other authors declare no competing interests.

## Additional information

**Article**

[1]Department of Nuclear Medicine, University Medical Center Leipzig, University of Leipzig, Leipzig, Germany. [2]Integrated Research and Treatment Center Adiposity Diseases, University Medical Center Leipzig, Leipzig, Germany. [3]Department of Medical Imaging, Radboud University Medical Center Nijmegen, Nijmegen, The Netherlands. [4]School of Biochemistry and Immunology, Trinity College Dublin, Dublin, Ireland. [5]Department of Nuclear Medicine, Section Radiopharmacy, University Hospital Augsburg, Augsburg, Germany. [6]Department of Neuroradiopharmaceuticals, Institute of Radiopharmaceutical Cancer Research, Research Site Leipzig, Helmholtz-Zentrum Dresden-Rossendorf (HZDR), Leipzig, Germany. [7]The Lübeck Institute of Experimental Dermatology, University Medical Center Schleswig-Holstein, Lübeck, Germany. [8]Department of Neurology, BG University Hospital Bergmannsheil, Ruhr-University Bochum, Bochum, Germany. [9]Medical Department III, Endocrinology, Nephrology, Rheumatology, University Medical Center Leipzig, Leipzig, Germany. [10]Department of Psychosomatic Medicine and Psychotherapy, Integrated Research and Treatment Center Adiposity Diseases, Behavioral Medicine Research Unit, Leipzig, Germany. [11]Helmholtz Institute for Metabolic, Obesity and Vascular Research (HI-MAG) of the Helmholtz Zentrum München at the University of Leipzig and University Hospital Leipzig, University of Leipzig, Leipzig, Germany. ✉e-mail: swen.hesse@medizin.uni-leipzig.de

