## [Transparent Peer Review file · Communications Biology]

Cholinergic network modulation in disinhibited eating behavior

Corresponding Author: Professor Swen Hesse

This manuscript has been previously submitted at another journal. This document only contains information relating to versions considered at Communications Biology.

Version 0:

Reviewer comments:

Reviewer #1

(Remarks to the Author)

This manuscript by Hesse and colleagues describes simultaneous [18F]flubatine PET/MRI Imaging in people with obesity and normal weight controls under two conditions: rest and during visual display of salient food cues. The authors report an increase in [18F]Flubatine VT during cues in people with obesity and high disinhibited eating behavior and some fMRI network changes/differences. There are a number of issues in the design, reporting of results, and interpretation of analysis that call into question the conclusions made by the authors. These are addressed below.

The statistical model for the primary PET analysis result is unclear. Please clarify how participants were classified into study group. Please clarify how multiple comparisons were considered. Please clarify how post-hoc analyses were performed to identify which groups have significant changes in the visual condition and include the statistics in the text of the results section. Please include the statistical result in the results, not the figure caption. Please check the stated degrees of freedom in the F statistic for accuracy. Please indicate statistically significant findings on Figure 1b. Please clarify if Figure 1c is a group average or representative image. Please double check the colorings of grids in Ext Figure 1.

The mechanistic rationale for increase in $\alpha 4\beta 2$ nAChR during visual stimulus is lacking. How would stimulation/recruitment of cortical circuits increase receptor availability? Receptor number does not typically change that dramatically on the time scales in question; alternatively an increase in availability could imply depletion of extracellular ACh, but this mechanism also seems inconsistent with stimulation of cortical circuits. These are PET data, not fMRI data.

General - what was the rationale for the identified ROIs? Why was clearance correction not applied to all extracted ROIs? Was clearance rate estimated from tissue or plasma curves, and was this rate different between groups? The abstract also does not contain any quantitative results, which is a major weakness. Are there 36 participants with obesity or 36 total participants?

The fmri/PET analysis is wholly inadequate, its not clear what hypothesis is being tested or why correlated $\alpha 4\beta 2$ nAChR availability would be expected or what that would mean. No relationship between regions defined by analysis of the obese group would be expected in the control group; the lack of relationship in controls shown in Figure 3a is expected and does not strengthen the meaning of the finding.

Reviewer #2

(Remarks to the Author)

This paper investigates the role of $\alpha 4\beta 2$ nicotinic acetylcholine receptor (nAChR) in the context of obesity and disinhibited eating, using simultaneous PET-fMRI to explore receptor availability and neural network connectivity in vivo in humans. While the study is conceptually compelling, I have a few comments/suggestions.

1. The sample size seems to be on the lower end for statistical power, especially for subgroup analysis. Please clarify.

2. It is noticeable that while statistical significance is set at 0.05, visual cue stimulation (line 139) for VTA is indicated as * for 0.089. Please clarify and correct it.

3. The reviewer is curious if visual food cues always equate to salience? Did the authors consider bright, colorful, contrasting cues or size and shape? Some details regarding those aspects of the methods could be useful. The cue design could benefit from details such as: were the food image individualized or standardized? Were satiety states controlled such as time since last meal?

4. There can be many alternative biological reasons behind receptor availability such as trafficking, desensitization and compensatory upregulation. This should be at least discussed in the discussion.

Reviewer #3

(Remarks to the Author)

I commend the authors on this valuable addition to the understanding of disinhibited eating behavior. This nuanced and in depth analysis of human imaging helps improve understanding of differences in human behavior.

Version 1:

Reviewer comments:

Reviewer #2

(Remarks to the Author)

All comments have been addressed.

Referees' Comments to Author and Reply (in italics):

Reviewer #1

Comments to the Author

This manuscript by Hesse and colleagues describes simultaneous [¹⁸F]flubatine PET/MRI Imaging in people with obesity and normal weight controls under two conditions: rest and during visual display of salient food cues. The authors report an increase in [¹⁸F]Flubatine VT during cues in people with obesity and high disinhibited eating behavior and some fMRI network changes/differences. There are a number of issues in the design, reporting of results, and interpretation of analysis that call into question the conclusions made by the authors. These are addressed below.

The statistical model for the primary PET analysis result is unclear. Please clarify how participants were classified into study group. Please clarify how multiple comparisons were considered. Please clarify how post-hoc analyses were performed to identify which groups have significant changes in the visual condition and include the statistics in the text of the results section. Please include the statistical result in the results, not the figure caption. Please check the stated degrees of freedom in the F statistic for accuracy. Please indicate statistically significant findings on Figure 1b. Please clarify if Figure 1c is a group average or representative image. Please double check the colorings of grids in Ext Figure 1.

Response: We sincerely thank the reviewer for his feedback and apologize for the somewhat confusing presentation of the results of the primary PET analysis. We proceeded as follows in the analyses: We first applied a mixed ANOVA to assess between-subjects effects, which included normal weight participants and participants with obesity, regardless of whether they exhibited high or low disinhibited eating behavior. This analysis yielded significant results for the VTA and the NBM also after testing for multiple comparisons as indicated below (Table). The two-way mixed ANOVA excludes the pairs of values with missing measures. This also explains the differences in the degrees of freedom/F-statistics between this analysis and the second analysis where we tested between-group effects (ANCOVA with age as covariate) comparing normal-weight vs. obese for the resting condition and for the stimulus condition considering whether participants with obesity had high or low disinhibited eating behavior. This analysis revealed the predicted higher thalamic $\alpha4\beta2^$ nAChR*

availability under stimulation in individuals with obesity high-disinhibited eating behavior compared with normal-weight healthy individuals. We also tested within each group/subgroup the distribution volumes V_T under rest versus under stimulus condition depicting higher V_T under stimulation in individuals with obesity and high-disinhibited eating behavior with borderline significance. This is now included in the text (tracked-changes-version, pages 5-6, lines 117-146) and figure caption was cleared.

Table: mixed ANOVA to assess between- and within-subjects effects (NW, normal-weight; OB; obesity)

Region	Between (NW vs. OB)	Post-hoc	Within (rest vs. stimulation)	Post-hoc	Interaction
Amygdala	F(1,25) = 0.017 p = 0.896		F(1,25) = 0.039 p = 0.845		F(1,25) = 0.975 p = 0.333
Hypothalamus	F(1,25) = 0.097 p = 0.758		F(1,25) = 0.0008 p = 0.978		F(1,25) = 0.385 p = 0.541
Insula	F(1,25) = 0.429 p = 0.519		F(1,25) = 0.089 p = 0.768		F(1,25) = 1.684 p = 0.206
NAc	F(1,25) = 0.164 p = 0.689		F(1,25) = 0.008 p = 0.931		F(1,25) = 0.833 p = 0.37
PFC	F(1,25) = 0.336 p = 0.567		F(1,25) = 0.133 p = 0.718		F(1,25) = 0.806 p = 0.378
Thalamus	F(1,25) = 0.741 p = 0.398		F(1,25) = 0.145 p = 0.706		F(1,25) = 2.294 p = 0.142
VTA	F(1,25) = 5.18 p = 0.032	p = 0.00647	F(1,25) = 0.229 p = 0.636		F(1,25) = 1.347 p = 0.257
NBM	F(1,25) = 5.99 p = 0.022	p = 0.00037	F(1,25) = 0.002 p = 0.968		F(1,25) = 0.594 p = 0.448
Cerebellum	F(1,25) = 1.244 p = 0.275		F(1,25) = 0.013 p = 0.909		F(1,25) = 1.947 p = 0.175

We also apologize the lack of labeling of statistically significant findings in the Figure 1b, which was added in the revised version of the manuscript (page 9, line 153), and for the misleading description of Figure 1c which display a representing image of a single case. This is clarified in the revised version of the manuscript (page 12, lines 180-183). We also add an image of a group average showing the differences between rest and stimulation in the three different cohorts (page 10, line 156, and page 12, lines 184-187).

We thank the review for pointing out the colorings of the grids. We did not always use coloring for high correlation coefficients. Instead, we only colored for significant correlations only. The grids of the expanded Figure 1F were not colored, as the small cohort size did not result in a significant p-value (>0.05) despite high correlation coefficients. This notion was added in the legend of the Figure (page 39, lines 983-98). The data are FDR-corrected, as indicated in the caption (page 39, line 980).

The mechanistic rationale for increase in $\alpha 4\beta 2$ nAChR during visual stimulus is lacking. How would stimulation/recruitment of cortical circuits increase receptor availability? Receptor number does not typically change that dramatically on the time scales in question; alternatively an increase in availability could imply depletion of extracellular ACh, but this mechanism also seems inconsistent with stimulation of cortical circuits. These are PET data, not fMRI data.

Response: We thank the reviewer for this important reference, which points to the importance of $\alpha 4\beta 2$ * nAChR in attentional control [1]. This include stabilization and performance recovering in the presence of a distractor (i.e., salient food cues), and augmented levels of tonic cholinergic activity as the main mechanism. Although we were unable to directly measure tonic cholinergic activity versus cholinergic transients, we proposed a model of altered cholinergic transmission in obesity, specifically with low top-down control (i.e., disinhibition) that includes the maintenance of attentional control in normal-weight healthy controls and poor attentional control in people with obesity and high-disinhibited eating behavior based on preclinical and modeling data [2,3]. Thus, on receptor dynamics level, cholinergic signaling enhances the ability to discriminate between irrelevant and relevant sensory input through stimulation of nAChR, which enhances afferent thalamocortical signaling, whereas it decreases cortical excitatory feedback connections and internal processing, as these processes would interfere with upcoming new information about environmental stimuli [1,4-6].

We found a stable PET signal in normal-weight controls between rest and stimulus while in participants with obesity lower $\alpha 4\beta 2$ * nAChR availability compared with normal-weight controls was

detected under rest and an increase (in the thalamus), or at least a tendency to, higher $\alpha 4\beta 2^*$ nAChR was found in obesity with disinhibited eating behavior probably indicating abnormal cholinergic transients accumulating to higher cholinergic tone when rewarding cues are presented. As such, with regard to the time-scale, we considered that cholinergic network modulation acts on a rather fast, transient (phasic) responses within a millisecond timeframe, as well as slower, longer lasting (tonic) modulatory effects [3,7,8]. The phasic release of ACh allows for the detection of the cue, while the longer lasting, tonic increase of ACh release represents a higher neuromodulation tone to enhance attentional performance [9]. According to this microdialysis study, tonic increase of ACh release compared to baseline enhances attentional performance on a scale of a few (~10) minutes through activation of thalamocortical circuits, amplifying phasic ACh release. Optogenetic stimulation of cholinergic transients in mice in a stimulus-response paradigm even increases invalid reporting of cues due to artificial dysregulation of phasic cholinergic signaling [10] while tonic release profile with changes in cholinergic activity that occur over minutes with higher tonic levels predicting greater amplitudes of phasic signals and enhanced cue detection [11] underscoring the understanding of cholinergic signaling as “arousal”-promoting. These changes in tonic activity was addressed by our study design of a bolus-infusion protocol allowing to detect changes at a receptor level [12-14].

In contrast to pharmacological challenge studies where downregulation is observed, we found higher availability during stimulation and attribute these changes primarily to the allosteric nature of the receptor type with more available binding sites in an activated sensitive open state (following a desensitized state after ligand binding and receptor activation)[3]. However, we fully agree with the reviewer that such hypothesized mechanism of action and surge for ligands is far from being proven by the data since ligand affinity is highest in the desensitized nAChR state compared to the activated state, the nAChR can only exert its excitatory function in the open state [8, 15]. Hence, depletion of ACh could also be discussed but, as mentioned by the reviewer, does not explain cortical activity under stimulation. Taken together, we now introduce this uncertainty in the explanation of the findings of our study to the discussion part of a mechanistic rationale on page 19, lines 305-311.

[1] Sarter M, Paolone G. Deficits in attentional control: cholinergic mechanisms and circuitry-based treatment approaches Behav Neurosci 2011; 125: 825-35. [2] Paolone G, et al. Cholinergic control over attention in rats prone to attribute incentive salience to reward cues. J Neurosci 2013; 33: 8321-35. [3] Graupner M, Gutkin B. Modeling nicotinic neuromodulation from global functional and network levels to nAChR based mechanisms. Acta Pharmacol Sin 2009; 30: 681-93. [4] Hasselmo ME, McGaughy J. High acetylcholine levels set circuit dynamics for attention and encoding and low acetylcholine levels set dynamics for consolidation. Prog Brain Res 2004; 145: 207-31. [5] Thiele A, Bellgrove MA. Neuromodulation of Attention. Neuron 2018; 97: 769-85. [6] Ding, J. B., et al. Thalamic gating of corticostriatal signaling by cholinergic interneurons. Neuron 2010; 67: 294-307. [7] Ballinger EC, et al. Basal forebrain cholinergic circuits and signaling in cognition and cognitive Decline. Neuron 2016; 91: 1199-1218. [8] Parikh V, Sarter M. Cholinergic mediation of attention: Contributions of phasic and tonic increases in prefrontal cholinergic activity. in Annals of the New York Academy of Sciences 2008; 1129: 225-35. [9] Sarter M, et al. Phasic acetylcholine release and the volume transmission hypothesis: time to move on. Nat Rev Neurosci 2009; 10: 383-90. [10] Gritton HJ, et al. Cortical cholinergic signaling controls the detection of cues. Proc Natl Acad Sci USA 2016; 113: E1089-97. [11] Parikh V, et al. Prefrontal Ach release controls cue detection on multiple time scales. Neuron 2007; 56 141-54. [12] Hillmer AT, et al. Imaging of cerebral $\alpha 4\beta 2^*$ nicotinic acetylcholine receptors with (-)-[¹⁸F]Flubatine PET: Implementation of bolus plus constant infusion and sensitivity to acetylcholine in human brain. Neuroimage 2016; 141: 71–80. [13] Bhatt S, et al. Evaluation of (-)-[¹⁸F]Flubatine-specific binding: Implications for reference region approaches. Synapse 2018; 72: 10. [14] Hillmer AT, Carson RE. Quantification of PET infusion studies without true equilibrium: A tissue clearance correction. J Cereb Blood Flow Metab 2020; 40: 860-74; Smart K, PET Imaging Estimates of Regional Acetylcholine Concentration Variation in Living Human Brain. Cereb Cortex 2021; 31: 2787-98. [15] Picciotto MR, et al. It's not “either/or”: activation and desensitization of nicotinic acetylcholine receptors both contribute to behaviors related to nicotine addiction and mood. Prog Neurobiol 2008; 84: 329-42

General - what was the rationale for the identified ROIs? Why was clearance correction not applied to all extracted ROIs? Was clearance rate estimated from tissue or plasma curves, and was this rate different between groups? The abstract also does not contain any quantitative results, which is a major weakness. Are there 36 participants with obesity or 36 total participants?

Response: We thank the reviewer for this comment on the selections of ROIs, which was not as clearly stated in the manuscript. ROI were extracted according to their role in central cholinergic transmission, either as source regions or downstream with important modulatory function within the attentional and reward circuits and networks. These regions include the amygdala as central to the modulation of emotional and affective state [16], the hypothalamus for the regulation of body homeostasis, critically modulated by cholinergic signaling [17]; the insula for processing salient cues

as well as for interoceptive awareness [18], the NAc as a key region in mesolimbic reward signaling modulated by the cholinergic system [19-20], the nucleus basalis of Meynert as the source region for cholinergic innervation of almost the entire cortex mantle [21]; the PFC for the modulation of food-cue reactivity [22], the thalamus with the highest amount of nAChR [23] and a crucial role in processing of sensory information as well as action-selection circuitry involving cholinergic neuromodulation responding to external cues [5], and the VTA which plays a central role next to NAc in mesolimbic reward signaling modulated by cholinergic neurons from the pedunculo-pontine-lateral dorsal tegmental nuclei [24], which themselves were not delineated in this analysis. According to our hypothesis with regard to the PET outcome measures, we focused on the thalamus while analyses of the cortical and limbic regions were exploratory. We have now added the criteria for ROI selection in the Methods part on page 31, lines 727-730.

Applying (-)-[¹⁸F]flubatine among other a reversibly binding radioligands, Hillmer and Carson investigated whether constant infusion to establish true equilibrium for quantification may lead to bias in outcome measurements if this assumption is not met. Their work developed and validated a correction that reduces bias in V_T estimates when true equilibrium is not present but this is only true for the thalamus with its very high V_T [14]. We published the time-activity-curve achieved in our cohort recently [25] (figure) showing that this was also not the case for the VTA and the NBM, regions that were not analysed in the work by Hillmer and Carson.

According to [14], we estimated clearance rate from tissue curves in all of the above mentioned regions; which did not revealed significant changes in the the outcome measures after FDR correction between the groups ($p_{FDR} > 0.05$). In detail, clearance rate β were for the thalamus (normal-weight -0.00293 ± 0.00002 versus obesity -0.00291 ± 0.00018 , $p = 0.61$, $p_{FDR} = 0.61$), for the VTA (-0.00296 ± 0.00002 versus -0.00299 ± 0.00008 ; $p = 0.04$; $p_{FDR} = 0.12$), and for the NBM (-0.00302 ± 0.00003 versus -0.00304 ± 0.00008 , $p = 0.18$; $p_{FDR} = 0.27$).

We agree with the reviewer that the results of the study are part of the abstract. However, we have adapted the abstract to the format of the journal *Communication Biology*, which, similar to other Nature Publishing Group journals does not regularly include this material. We have now included the

number of the participants studies, which totaled 36, to emphasize the pilot nature of the study In consultation with the publisher/editors, we will also include other results where possible.

[16] Rasia-Filho AA, et al. Functional activities of the amygdala: An overview. *Journal of Psychiatry and Neuroscience* 2000, 25: 14-23. [17] Kenny PJ. Common cellular and molecular mechanisms in obesity and drug addiction. *Nat Rev Neurosci* 2011; 12: 638-51. [18] Menon V, et al. Saliency, switching, attention and control: a network model of insula function. *Brain Struct Funct* 2010; 214: 655-67. [19] Avena, N. M. & Rada, P. V. Cholinergic modulation of food and drug satiety and withdrawal. *Physiol Behav* 2012; 106, 332-36. [20] Shariff M, et al. Neuronal nicotinic acetylcholine receptor modulators reduce sugar intake. *PLoS One* 2016; 11: e0150270. [22] Picciotto MR, et al. Acetylcholine as a neuromodulator: Cholinergic signaling shapes nervous system function and behavior. *Neuron* 2012; 76: 116-29. [22] Kroemer NB, et al. Nicotine enhances modulation of food-cue reactivity by leptin and ghrelin in the ventromedial prefrontal cortex. *Addiction Biology* 2015; 20: 832-44. [23] Sabri O, et al. First-in-human PET quantification study of cerebral $\alpha 4\beta 2^*$ nicotinic acetylcholine receptors using the novel specific radioligand (-)-[^{18}F]Flubatine. *Neuroimage* 2015; 118, 199-208. [24] Avena NM, Rada PV. Cholinergic modulation of food and drug satiety and withdrawal. *Physiol Behav* 2012; 106: 332-6. [25] Schweickert de Palma E, et al. Availability of central $\alpha 4\beta 2^*$ nicotinic acetylcholine receptors in human obesity. *Brain Sci* 2022; 12: 1648.

The fmri/PET analysis is wholly inadequate, its not clear what hypothesis is being tested or why correlated $\beta 2^*$ nAChR availability would be expected or what that would mean. No relationship between regions defined by analysis of the obese group would be expected in the control group; the lack of relationship in controls shown in Figure 3a is expected and does not strengthen the meaning of the finding.

We sincerely thank the reviewer for discussing this key figure, which shows the relationship between individual changes in V_T and the strength of functional connectivity (beta estimates) derived from the network clusters comparing rest and stimulation in obesity. The aim of this analysis was to show that these parameters are closely correlated in study participants with obesity but not in normal-weight controls. We fully agree that the lack of relationship in the control group is expected; however, the relationship between V_T and beta estimates was not obvious based on the findings of PET or fMRI analyses alone. Instead, such voxel-based provide hypothesis-free 2nd analysis. Following the reviewers suggestion, we reanalyzed the data to provide clues for a relationship between the changes of V_T in the thalamus observed in obesity according to our hypothesis and replaced the figure by a new one that shows a strong correlation between V_T in the thalamus (obtained from VOI-analysis and the strength of functional connectivity in both masks. This is now explained in the text on page 14, lines 219-223) and shown in Fig. 3 (page 15, lines 233-234). We further processed the presentation of the joint fMRI/PET analysis by applying the mask from the rest-vs-stim studies in normal-weight controls to show that significant correlation is obtained in study participants with obesity, highlighting a cholinergic modulated switch in brain networks mediated by thalamic $\alpha 4\beta 2$ nAChR when salient food stimuli are presented (see also the revised text on page 14, lines 223-229). In addition, supplementing the previous Fig. 3 we performed a similar analysis applying the mask that was obtained when comparing rest-vs-stim fMRI in normal-weight controls to the entire sample indicating (as expected) no correlations between V_T and beta estimates. Both the previous Fig. 3 and the new generated Figure of voxel-based analyses were moved to the Supplementary material (page 44, lines 1010-1024). Of note, the correlation coefficient and the p-values in the former Figure 3 were corrected.

Reviewer #2

Comments to the Author

This paper investigates the role of $\alpha 4\beta 2$ nicotinic acetylcholine receptor (nAChR) in the context of obesity and disinhibited eating, using simultaneous PET-fMRI to explore receptor availability and neural network connectivity in vivo in humans. While the study is conceptually compelling, I have a few comments/suggestions.

1. The sample size seems to be on the lower end for statistical power, especially for subgroup analysis. Please clarify.

Response: *We sincerely thank the reviewer for discussing this point. The assumption that the sample size seems to be on the lower end for statistical power, especially for subgroup analysis is*

correct. The study was initiated as a pilot study for further hypothesis generation. The case number estimation is based on the results of an earlier receptor study assuming a mean value of 1.5 of the dimensionless receptor parameter binding potential (BP) in receptor-rich brain regions [1]. Since in the present study a PET/MR measurement of the binding potential BP (which is $V_T - 1$) was performed twice on one patient (without and after stimulus, matched-pair study), we expect that the parameter differences will be normally distributed with a standard deviation of 0.15 (assuming a test-retest accuracy of the PET/MR scanner of 10%) [2]. Therefore, in a paired t-test, by testing 10 patients, we will be able to reject the null hypothesis of a response difference of 0 with power 0.9 if the actual response difference is 0.17 or greater. The type 1 error probability associated with this test of the null hypothesis is $\alpha = 0.05$. Notably, The mentioned studies applied radiotracers different from (-)-[¹⁸F] flubatine, which itself appears to have higher V_T [3,4] in comparison with those.

However, with a conservative approach we were going to investigate 60 non-smoking volunteers in total: $n = 40$ have obesity with $BMI > 35 \text{ kg/m}^2$ of whom $n = 20$ have high disinhibition and $n = 20$ have low disinhibition and $n = 20$ are healthy controls with $BMI < 25 \text{ kg/m}^2$ and low disinhibition. Unfortunately, the study was interrupted by the COVID-19 pandemic, so we were finally able to reach the current number of study participants with all our resources, which was estimated to be low but just sufficient for statistical significance. Ultimately, we decided that despite this limitation, we found these novel and very encouraging results useful to disseminate for planning future research.

[1] Hesse S, et al. The serotonin transporter availability in untreated early-onset and late-onset patients with obsessive-compulsive disorder. *Int J Neuropsychopharmacol* 2011; 14: 606-17. [2] Kaller S, et al. Test-retest measurements of dopamine D1-type receptors using simultaneous PET/MRI imaging. *Eur J Nucl Med Mol Imaging* 2017; 44: 1025-32. [3] Sabri O, et al. First-in-human PET quantification study of cerebral $\alpha 4\beta 2^*$ nicotinic acetylcholine receptors using the novel specific radioligand (-)-[¹⁸F]Flubatine. *Neuroimage* 2015; 118: 199-208. [4] Hillmer AT, et al. Imaging of cerebral $\alpha 4\beta 2^*$ nicotinic acetylcholine receptors with (-)-[¹⁸F]Flubatine PET: Implementation of bolus plus constant infusion and sensitivity to acetylcholine in human brain. *Neuroimage* 2016; 141: 71-80.

2. It is noticeable that while statistical significance is set at 0.05, visual cue stimulation (line 139) for VTA is indicated as * for 0.089. Please clarify and correct it.

Response: We thank the reviewer for pointing this out. This was a mistake. Rather NBM should have been marked with this asterisk. We have corrected the typo in the revised version, which now also has a different presentation of the results according to the comments of reviewer #1. Thus, we transferred the figure caption to the text (tracked-changes-version, page 5, lines 117-146) and one figure indicating differences in the mean V_T maps in all the three groups on page 11, lines 156-157.

3. The reviewer is curious if visual food cues always equate to salience? Did the authors consider bright, colorful, contrasting cues or size and shape? Some details regarding those aspects of the methods could be useful. The cue design could benefit from details such as: were the food image individualized or standardized? Were satiety states controlled such as time since last meal?

Response: Thank you for this important comment regarding the design and salience of the visual food cues. The food images used in our study were standardized and created under strictly controlled photographic conditions at the Max Planck Institute for Human Cognitive and Brain Sciences in Leipzig, Germany. All foods were photographed on white plates using consistent lighting, camera angle, and distance to ensure uniform contrast, brightness, size, and shape presentation across stimuli.

To account for individual variation in food preferences and perceived salience, the full set of 180 food images was pre-evaluated by over 100 independent participants who rated each image along the dimensions of "tastiness" and "healthiness." Based on these normative ratings, food stimuli were categorized (e.g., high vs. low calorie; sweet vs. savory), and each participant in the main experiment was shown a balanced subset of images from each category. This approach ensured both perceptual control and psychological relevance while minimizing confounds related to visual salience alone.

While all images were standardized in presentation, we acknowledge that salience is not purely visual but also contextual and individual. Therefore, our design combines visual standardization with subjective relevance, providing a controlled yet ecologically valid set of cues. The stimulation material was previously published [3]. Additionally, we have added a set of visual stimuli on pages 47-48, lines 1037-1039 of the Supplementary material (Supplementary Fig. 6).

We also thank the reviewer for the comment regarding controlled meal standards, which was not detailed explained in the manuscript. Sorry for that. The individual scan day was the same in every study participant: Study participants were asked to have a light breakfast before arriving at the PET unit in the morning. The scan starts at 10:00 am and participants were asked immediately beforehand about their feelings of hunger and satiety by using the VAS, which did not differ between the group with mean values of 16.4 ± 15.8 in normal-weight controls, 19.3 ± 13.9 in individuals with obesity and low-disinhibited eating behavior and 15.7 ± 14.2 in individuals with obesity and high-disinhibited eating behavior ($p = 0.16$). We have now added this information to the individual study program on page 30, lines 701-708.

[3] Markman M, et al. Differences in Discounting Behavior and Brain Responses for Food and Money Reward. eNeuro 2024; 11: ENEURO.0153-23.2024

4. There can be many alternative biological reasons behind receptor availability such as trafficking, desensitization and compensatory upregulation. This should be at least discussed in the discussion

Response: We are very thankful for this comment since this address probably the major point of discussion that cannot be clarified by in human data so far. Given that the pharmacological changes at the nAChR [4] is modulator type- and dose-dependent, we only consider the ability of Ach to coordinate the response of neuronal networks in many brain areas to ongoing stimuli that makes cholinergic modulation an essential mechanism underlying complex behaviors. In keeping with this we would like to propose a possible mechanism based on previous modeling data [5]. We added this notion at page 19, lines 305-311.

[4] Taly A, et al. Nicotinic receptors: allosteric transitions and therapeutic targets in the nervous system. Nat Rev Drug Discov 2009; 8: 733-50. [5] Graupner M, Gutkin B. Modeling nicotinic neuromodulation from global functional and network levels to nAChR based mechanisms. Acta Pharmacol Sin 2009; 30: 681-93.

Reviewer #3

I commend the authors on this valuable addition to the understanding of disinhibited eating behavior. This nuanced and in depth analysis of human imaging helps improve understanding of differences in human behavior.

Response: We sincerely thank the expert for this positive assessment and encouraging feedback. We hope that we can expand the data with future studies and that it will be further replicated.